# Clinicians’ Views and Experiences with Offering and Returning Results from Exome Sequencing to Parents of Infants with Hearing Loss

**DOI:** 10.3390/jcm11010035

**Published:** 2021-12-22

**Authors:** Lauren Notini, Clara L. Gaff, Julian Savulescu, Danya F. Vears

**Affiliations:** 1Melbourne Law School, University of Melbourne, Carlton, Melbourne 3052, Australia; lauren.notini@monash.edu; 2Biomedical Ethics Research Group, Murdoch Children’s Research Institute, Parkville, Melbourne 3052, Australia; julian.savulescu@philosophy.ox.ac.uk; 3Genomics in Society, Murdoch Children’s Research Institute, Parkville, Melbourne 3052, Australia; clara.gaff@melbournegenomics.org.au; 4Department of Paediatrics, University of Melbourne, Parkville, Melbourne 3052, Australia; 5Melbourne Genomics Health Alliance, Parkville, Melbourne 3052, Australia; 6The Oxford Uehiro Centre for Practical Ethics, University of Oxford, Oxford OX1 4BH, UK; 7University of Melbourne, Parkville, Melbourne 3052, Australia; 8Centre for Biomedical Ethics and Law, KU Leuven, 3000 Leuven, Belgium

**Keywords:** genomic sequencing, hearing loss, newborn screening, bioethics

## Abstract

Exome sequencing (ES) is an effective method for identifying the genetic cause of hearing loss in infants diagnosed through newborn hearing screening programs. ES has the potential to be integrated into routine clinical care, yet little is known about the experiences of clinicians offering this test to families. To address this gap, clinicians involved in a clinical study using ES to identify the cause of infants’ hearing loss were interviewed to explore their experiences with offering and returning results to parents. Interview transcripts were analysed using inductive content analysis. Twelve clinicians participated: seven genetic counsellors, four clinical geneticists, and one paediatrician. Most clinicians were supportive of offering ES to infants with hearing loss, primarily because results may inform the child’s clinical management. However, some expressed concerns, questioning the utility of this information, particularly for isolated hearing loss. Clinicians had differing views regarding the optimal time to offer ES to families; while some felt that families can manage everything at once, others recommended delaying testing until parents have come to terms with their child’s diagnosis. These findings show the complexity involved in determining how ES should be offered to families following the diagnosis of a child with hearing loss, particularly with regards to when testing is suggested.

## 1. Introduction

Approximately 0.1% of Australians are born with unilateral or bilateral hearing loss [1]. Although ranging in severity from mild to profound, moderate to profound congenital hearing loss can usually be detected in the first few months of life via routine newborn hearing screening, which typically involves measuring Automated Auditory Brainstem Response (AABR). Congenital hearing loss may be genetic and several hundred genes have been linked to hearing loss [1]. Most genetic forms of hearing loss are isolated (with hearing loss being the only symptom), and the remaining (approximately 30%) are syndromic (where other symptoms are present). In syndromic cases, the other symptoms may not present until late childhood or adulthood (e.g., vision loss in Usher syndrome).

Research shows that exome sequencing (where the coding regions of a person’s DNA are sequenced) can be used to identify the genetic cause of hearing loss, with a higher diagnostic yield than standard single gene testing [2]. A clinical study, known as the Melbourne Genomics Congenital Deafness Project, offered exome sequencing to infants with hearing loss who were born from January 2016 to December 2017 in Victoria, Australia. The purpose of this study was to investigate the process and outcomes of providing exome sequencing within the clinical setting. To achieve this, an effectiveness–implementation hybrid approach was adopted, which gathered information on the delivery of the test and its potential for integration into clinical care [3,4]. Infants were eligible to participate if they received a diagnosis of permanent, bilateral congenital hearing loss (of moderate, severe, or profound severity) via the Victorian Infant Screening Program. Infants with unilateral, temporary, or mild hearing loss were excluded [1]. These criteria were selected based on the likely yield for identifying the genetic basis of the hearing loss, which was deemed much lower in unilateral and mild cases. Recruitment occurred at two tertiary paediatric hospitals located in Melbourne, Victoria, Australia, as well as affiliated regional clinics. As exome sequencing cannot identify all genetic causes of hearing loss, chromosome microarrays were also performed. Together, testing identified a genetic cause of hearing loss in 56% (59/106) of the infants who participated in the study, leaving 44% for whom a causative variant was not identified [2]. Parents were also offered the option to receive additional findings (findings unrelated to the rationale for testing but relating to childhood-onset conditions, both clinically and non-clinically actionable) [5]. Results were also returned to parents by genetic health professionals as per the clinical setting.

The study by Downie et al. showed that, based on the high diagnostic yield and clinical implications, exome sequencing following the identification of infants with hearing loss has the potential to be integrated into routine clinical care [2]. As such, it is important to understand the experiences and challenges faced by clinicians during the testing process. Although others have explored clinicians’ experiences with offering exome sequencing in other contexts [6,7,8], a diagnosis of hearing loss is unique in that infants are diagnosed very young (some under three months, but most between three and nine months) [2]. In order to contribute to addressing this gap, we aimed to interview health professionals who were offering exome sequencing to infants following a diagnosis of hearing loss to capture their views and experiences with the processes of offering and returning these results.

## 2. Materials and Methods

Clinicians involved in the overarching clinical study (including genetic counsellors, clinical geneticists, and paediatricians) were deemed eligible to take part in this qualitative interview study. Eligible clinicians, who were identified by the clinical project lead, were emailed an invitation to participate. In accordance with qualitative research methodology, participant recruitment continued until data saturation was reached (when minimal new data was generated addressing the study aims) and we had achieved sufficient sample heterogeneity by recruiting at least one clinician from each of the clinical specialties involved in the overarching clinical study (genetic counselling, clinical genetics, and paediatrics) and each of the study sites. Our sample size was not intended to be statistically representative, but rather to generate rich data that addressed our study aims [9,10]. The semi-structured interview guide included open-ended questions regarding the practical and ethical experiences and challenges related to the testing process as follows: (a) clinicians’ views on whether and when exome sequencing should be offered to parents of children with hearing loss; (b) clinicians’ accounts of parents’ reasons for and against exome sequencing for their child’s hearing loss; (c) clinicians’ experiences offering hearing loss-related exome sequencing and returning results to parents. We also asked about their experiences with offering additional findings to parents, the results of which are reported elsewhere (manuscript under review). The interview guide was developed by Lauren Notini and Danya Vears, and feedback was provided by the clinical study lead who is a clinical geneticist.

Interviews were conducted by Lauren Notini, either in person or by telephone depending on the location and preference of the participant. In-person interviews were conducted at the participant’s workplace, either in a meeting room or in the participant’s office. Each interview was audio-recorded and transcribed. Interview transcripts were analysed using inductive content analysis, a method of qualitative data analysis that involves using content categories generated from the data, rather than predetermined categories [11,12]. Coding continued iteratively until all data relevant to the research question had been coded into categories and subcategories. All transcripts were coded by Lauren Notini, and two of the transcripts were also coded by Danya Vears to verify the coding scheme. Differences in coding were identified and resolved via discussion. To protect participants’ confidentiality, pseudonyms are used and participants’ demographic information is reported collectively.

## 3. Results

### 3.1. Participant Characteristics

A total of 12 clinicians participated, including seven (58%) genetic counsellors, four (33%) clinical geneticists, and one (8%) paediatrician; this was more than half of the 21 clinicians eligible to participate in the study. Average interview length was 41 min, with a range between 25–59 min. Clinicians varied in terms of how many years they had worked in their current profession; experience ranged between trainee level (*n* = 1) and >20 years (*n* = 1) with the majority of participants having from 1–4 years (*n* = 4) or 5–9 years (*n* = 3) experience. While clinicians were recruited from both study sites, the majority (10/12, 83%) worked at one site. This uneven distribution of specialties and worksites among our sample precludes meaningful comparison of participants’ responses. Hence, we do not provide separate analyses of the views expressed by clinicians at different worksites, nor provide a comparative analysis of responses from the different clinical specialties.

### 3.2. Clinicians’ Experiences with Offering Exome Sequencing for Hearing Loss

Three main content categories were generated from the interview data: (1) clinicians’ views on offering exome sequencing to parents of children with hearing loss; (2) clinicians’ accounts of parents’ decision making regarding exome sequencing for their child’s hearing loss; (3) clinicians’ experiences returning diagnostic exome sequencing results to parents. We describe these content categories, providing illustrative quotes for each, below. Emphasis in quotes represents participants’ original emphasis.

#### 3.2.1. Clinicians’ Views on Offering Exome Sequencing for Hearing Loss

Most clinicians were supportive of offering exome sequencing to parents of children with hearing loss, for several reasons. This was often because the results of exome sequencing may inform the child’s clinical management:


*“The diagnosis of certain genetic syndromes will lead you down a different treatment pathway and allow you to avoid other investigations.”*
(Clinician 1)

Some clinicians also put forward reasons relating to equity, and felt that exome sequencing should be offered to parents of children with any type of hearing loss, and not just the types that met the study eligibility criteria:


*“I think definitely the hearing component, everybody should be offered, or given the chance to take it up … even the unilateral losses and the mild losses which are not part of eligibility … I think they should also be offered the same testing because it’s just being equitable, they also deserve to have an answer, if there is one.”*
(Clinician 2)

Other reasons given by participants in support of offering exome sequencing to parents of children with hearing loss included that knowledge is power, the results may be helpful in addressing parents’ concerns, and that exome sequencing for hearing loss was seen as a cost-effective intervention worthy of resources.

Some clinicians also expressed concerns about offering exome sequencing for paediatric hearing loss. Several questioned the utility of this information, especially for isolated hearing loss, or what are the appropriate limits of parents’ abilities to make decisions for their children. Others queried whether hearing loss is better understood as a disability or a culture, and raised concerns relating to data storage:


*“It’s difficult with deafness, because deafness alone, we think of deafhood and people that are deaf live a great, long, healthy life. And there’s a bit of an ethical discussion to be had for testing newborn babies that are found to have hearing loss … to an extent you’re really pathologising this condition they have … these parents are making decisions for them and their genetic information is going to be stored indefinitely… when maybe there was no real indication for doing the test in the first place.”*
(Clinician 8)

Other clinicians raised ethical concerns relating to distributive justice, as some families had the opportunity to receive free exome sequencing via the study but other families who could also benefit from testing did not receive this same opportunity:


*“My main ethical challenge with this project is that it’s come to an end, and now we have lots of families coming to our clinic, and being referred, expecting this level of testing… this was a temporary solution, which was great, but now it’s stopped, we’re back to square one with a lot of these families and there’s a lot of deaf babies born every year and now I just have to sit there and say ‘it’s (AUD) 3000 dollars and we have no means of funding it whatsoever’ … So it’s a question of distributive justice that I am bothered by a lot.”*
(Clinician 4)

Clinicians had different views regarding the optimal time to offer exome sequencing to parents of children with hearing loss. Some clinicians felt that some families can manage everything at once:


*“There will be keen families, and they’ve got the psychological and coping resource to deal with everything at the same time. And they could come through early.”*
(Clinician 10)

However, most clinicians felt that exome sequencing should be offered to parents at a later date, often because the children were very young infants and parents were still coming to terms with their child’s hearing loss diagnosis. 


*“It was pretty clear that for some parents it was a bit early to be raising this. And so, if I was doing this more broadly, I would say maybe not to bring in the genetics and the genetic question ‘til six months of age or nine months or something like that. ‘Cause it did seem that in this cohort, keeping in mind they’ve just been diagnosed with deafness, so they were coping with that diagnosis, then to be faced with this offer of more stuff, I think it was probably a bit overwhelming for some people… I would say maybe we’ll just try and get these kids at six months, for example, rather than as early as possible.”*
(Clinician 1)

Families are also being invited to participate in other studies relating to hearing loss at this time, and some clinicians discussed the discomfort they and their colleagues experienced offering the test.


*“That feeling of, you know, are we bombarding the families with so much, at that particular point in their sort of understanding of what’s happening to their kids? So, I think people (study clinicians) felt a bit uncomfortable about that, that it was all happening at the wrong time … not the genetic testing for the cause for the deafness, but the offer, and the idea that they were being invited for two or three projects … simultaneously at a point when they’re still coming to grips with the diagnosis in the child.”*
(Clinician 10)

Some clinicians felt that the decision regarding exome sequencing has important implications, and, therefore, parents should be given more time to decide. Some clinicians felt that there should be no time limit on the offer, and parents should be able to opt in to receive exome sequencing at any time: 


*“Parents should have the option to opt in at any time to have it done, when they’re ready or when they have thought enough about it. ‘Cause there are huge implications once you’ve taken the test, in terms of knowing the result.”*
(Clinician 2)

#### 3.2.2. Parents’ Reasons for and against Project Participation

According to clinicians, the most common reason parents accepted exome sequencing for their child was to try to find out the cause of their child’s hearing loss, as this may help them to better understand their child’s prognosis and inform clinical management:


*“They want to know the answer because they want an explanation, or they want treatment.”*
(Clinician 11)

Clinicians reported that parents were particularly interested in establishing whether their child’s hearing loss was isolated or part of a syndrome:


*“… there’s a lot of information out there on the internet and within the groups as well … ‘now that we’ve found this one thing, could it be that there are other things that are going to be going on?’ And I think that for a lot of parents, they’re quite scared of conditions like Usher syndrome and those sorts of things, so I think that was a big motivation for a lot of parents to have the testing for hearing loss.”*
(Clinician 9)

Some clinicians also noted that obtaining a genetic diagnosis for their child’s hearing loss may help to alleviate any feelings of parental blame or guilt: 


*“Parents might feel guilty about something that’s happened in pregnancy… a genetic diagnosis for them would help take away that kind of guilt.”*
(Clinician 3)

Most clinicians reported that some parents also accepted exome sequencing for their child for reasons relating to reproductive planning. For example, some parents wanted to be better prepared if they had another child with hearing loss as noted by this clinician:


*“Not necessarily that they would do anything to prevent a pregnancy or terminate a pregnancy based on the result, but just so they had an idea of ‘is this likely to be a one-off thing? Could I have another child with this same hearing loss? Will it be a similar type?’ Just to know for that kind of family planning … a lot of families described finding out about the diagnosis as just such an earth-shattering shock to their system… I think they wanted to maybe be a bit more prepared if it happened again.”*
(Clinician 9)

In contrast, other parents wanted to use the results in order to avoid having another child with hearing loss:


*“They’re having more children and they wish to avoid having a child with hearing loss.”*
(Clinician 10)

Clinicians reported that some parents agreed to sequencing because they wanted to confirm the hearing loss diagnosis in their child or know more about their child’s health, including one family with a known genetic cause of hearing loss who chose to participate in the project to receive additional findings. Others wanted to participate to help explain the parents’ own hearing loss, or simply agreed because exome sequencing was offered to them as a clinical test.

Clinicians reported that parents had two main reasons for declining exome sequencing for their child’s hearing loss. First, clinicians reported that some parents declined participation as they felt that the exome sequencing results would not change anything: 


*“I just remember them saying, ‘I don’t mind whether my baby’s hearing loss is because of a genetic change or not, it’s not going to change anything for us or our family.’”*
(Clinician 6)

Second, other families declined participation as they were already overwhelmed by their child’s hearing loss diagnosis and management:


*“There’s so much support, programs, intervention related to hearing loss that the families just felt run off their feet and overwhelmed by all the services … they were like, ‘I can’t introduce another health service into the picture.’”*
(Clinician 11)

Clinicians explained that some parents declined exome sequencing because they already knew or suspected the cause of their child’s hearing loss or had other family members with hearing loss who felt they were coping well and, therefore, did not feel the need for further testing. Some parents felt that, given the possibility a diagnosis for their child’s hearing loss would not be identified, accepting testing might lead to disappointment. Others either did not want to be involved in research or had concerns about the potential implications of the findings, such as the impact of exome sequencing results on their child’s future ability to obtain insurance or concern it would reveal the child’s father was not their genetic father.


*“I think at the moment I’d just say it’s a mixture of different factors, which range from they may already have a genetic diagnosis, they may have decided the condition’s not genetic. They may just not want to be involved in research, or be concerned about insurance or other implications. So we’ve got a list, but there’s no sort of single factor that’s repeating, it’s basically a whole lot of different things.”*
(Clinician 1)

#### 3.2.3. Return of Exome Sequencing Results

At the time of interview, all clinicians had been involved in discussing exome sequencing results relating to hearing loss to parents. In describing their experiences returning these results, some clinicians reported that they had to remind parents about the testing that had taken place, as receiving the results was not always at the forefront of parents’ minds:


*“Usually the time between consent and results’ return was a number of months. So often when we were calling families to arrange a result appointment, you’d have to kind of re-introduce yourself and go ‘remember me? We talked about this test, the reason that I’m calling is we’re expecting to have some results for you in the next couple of weeks’. But I think a lot of the time you were kind of reminding families what exactly it was you were calling about ... I guess the sense of urgency (around the time of the diagnosis) was usually off by that stage.”*
(Clinician 6)

Some clinicians also found it challenging to explain variants of uncertain significance (i.e., genetic changes where there is currently insufficient evidence to determine whether or not it is the genetic cause) to families:


*“One thing we’ve found is that there seems to be … lots of variants (of uncertain significance) in hearing loss genes. So, it’s quite common to get a report with three or four variants of unknown significance. Or even one that we’re quite confident is the diagnosis and some variants of unknown significance. And I found that very challenging for some families, to explain a genetic finding and then explain other genetic findings that may be less relevant.”*
(Clinician 3)

Clinicians reported that parents who received an answer for their child’s hearing loss (genetic or otherwise) typically reacted positively, even though this information may not necessarily affect clinical management:


*“For the deafness results, I’ve been surprised at how many families have been really emotionally affected by giving them a deafness result … quite a few families are really ecstatic to have an answer, which I think maybe we didn’t predict at the start because it often doesn’t affect management or treatment that much … just really relieved to have an answer to explain what’s going on…that sense of, yeah, relief I guess is the best way to describe it. So that stood out.”*
(Clinician 3)

Some clinicians described this sense of relief as particularly strong in cases where parents had received a genetic explanation for their child’s hearing loss:


*“…often people blame themselves for problems in their children, so finding a genetic diagnosis brings a relief to some families that it’s not something that they did during the pregnancy, particularly when people are worried about an infection or something like that, that’s playing on their mind.”*
(Clinician 4)

According to participants, parents’ reactions to the exome sequencing result also depended on whether their child’s hearing loss was isolated or syndromic. Parents who received a molecular diagnosis of isolated (non-syndromic) hearing loss often expressed gratitude and relief to clinicians:


*“Relief was a big one. Just being grateful that they have an answer, that it doesn’t mean anything else is going to necessarily go wrong.”*
(Clinician 9)

However, other parents whose results confirmed an isolated hearing loss experienced negative feelings. Some held concerns about stigmatisation of genetic conditions or the impact of the results on future reproductive planning:


*“A subsample … were expecting it to come back genetic, but when it did it kind of just confirmed like ‘oh, it’s something that’s genetic.’ And I think maybe, like there’s still a stigma around genetic conditions and having something different with you. So, I think for some of them they may have been a bit taken aback, like ‘oh ok, it means there’s a 25% chance it could happen again if it was a recessive cause … not that I’d terminate a pregnancy based on it, but now I’m in a bit of a conflict of, there’s a chance I could have another child with a hearing loss’.”*
(Clinician 9)

For others, the result may have confirmed a hearing loss diagnosis that they were denying:


*“Humans are unpredictable (laughs). So, you think someone might be very happy that they’ve received an answer because it’s confirmed that it’s isolated hearing loss. The parent absolutely knew the child has hearing loss, and that, yep, they have the answer, they want to use it for reproductive choices, it’s ticking all those boxes for them. But in actual fact maybe they had somewhere in the back of their mind that they hoped it wasn’t really hearing loss and it was something the child will grow out of, and now you’ve just given them this in-writing thing that says ‘no, you’re child definitely has this’. And so for them it’s devastating.”*
(Clinician 5)

Some parents who received a syndromic diagnosis for their child’s hearing loss reacted relatively positively to this news because it allowed them to be proactive in their child’s clinical management:


*“For the syndromic forms of hearing loss … for some families they found it as a helpful explanation for things and appreciated that there could be other screening or other testing that could be done for their child, and were grateful that this was also picked up early… like ‘we found out they could have a heart condition, so we’re going to send them to have their heart looked at as well. It’s something we wouldn’t necessarily have done’ or ‘their kidneys looked at’. So, I think there was value in that for quite a lot of families.”*
(Clinician 9)

However, some clinicians mentioned parents who had received a syndromic diagnosis and reacted more negatively due to the devastating nature of the diagnosis or because this diagnosis was unexpected and meant extra medical management for their child: 


*“I was personally involved with a family who received a diagnosis of Usher syndrome, and that was just absolutely heartbreaking for them. It was their biggest fear, and it eventuated. So, yeah. It was utterly devastating for them.”*
(Clinician 9)

For parents who did not receive an explanation for their child’s hearing loss, clinicians reported mixed reactions. Many parents who did not receive an explanation reacted quite well and were particularly relieved that no syndromic diagnosis was found:


*“As much as we didn’t find a diagnosis at all, a lot of families were relieved that we didn’t find a syndromic diagnosis or something like that.”*
(Clinician 6)

Other parents who did not receive an answer for their child’s hearing loss reacted more negatively. This included parents who were wishing to use this result to inform decisions about future medical management and future reproductive planning:


*“…some families were quite disappointed if they didn’t get an answer, particularly if they were hoping to understand if their child needed any other check-ups moving forward other than just getting their hearing checked every now and again, or some families were keen to use that information in a reproductive setting for future pregnancies.”*
(Clinician 6)

Clinicians reported that future re-analysis of exome sequencing results was offered to these parents, many of whom expressed interest in this: 


*“People are interested in the possibility of re-analysis down the track, so most people expressed an interest in that when we brought it up as a possibility.”*
(Clinician 4)

## 4. Discussion

This is the first published study exploring clinicians’ views and experiences with offering and returning results from exome sequencing for hearing loss in infancy. Most of our participants highlighted the potential benefits of offering exome sequencing for infant hearing loss, such as informing the child’s clinical management. However, concerns about offering such testing were also raised by some clinicians, particularly in the context of isolated (non-syndromic) hearing loss, for which the clinical benefits of offering such testing may be less clear. Those who were in favour of testing cited reasons relating to clinical and personal utility, equity, and cost-effectiveness. Those who raised concerns also mentioned factors relating to utility, yet, in addition, questioned the boundaries of parental autonomy, deafness as a disability, and distributive justice. Interestingly, there was no clear pattern with clinician type, study site, or years of experience as to whether participants were more or less in favour of offering exome sequencing for hearing loss.

Despite the general agreement that exome sequencing for paediatric hearing loss should be offered to parents, clinicians had more varied views regarding the most optimal timing to offer such testing. While some clinicians felt some parents coped well with early testing, most felt testing should be offered later, once parents had had more time to come to terms with their child’s hearing loss diagnosis. This finding was supported by participants’ reports that some parents declined the offer because of how overwhelmed they were at the time. There are several advantages to offering exome sequencing to parents soon after the child’s diagnosis of hearing loss. First, exome sequencing increases diagnostic yield, which subsequently improves the quality of care infants receive, either through reducing the need for, or allowing for furthermore targeted further diagnostic investigations [2]. The cost-effectiveness of such testing has also been demonstrated based on these factors [13]. Second, parents who are hoping to receive an answer for their child’s hearing loss may obtain benefit, such as through reduced anxiety, if they can receive this information earlier in the diagnostic journey. Third, an earlier syndromic hearing loss diagnosis could allow parents more time to come to terms with this diagnosis and seek appropriate care for their child, such as additional monitoring, as well as engage in reproductive planning for future children.

However, there are also disadvantages to offering exome sequencing during early infancy. Parents may already be overwhelmed by either the fact they have a newborn or the recent hearing loss diagnosis. Offering exome sequencing at a later time could provide parents with more space to consider the various factors when making their decision regarding exome sequencing for hearing loss. The appropriate timing to offer exome sequencing for hearing loss will likely depend on the particular family and their circumstances. While offering early exome sequencing may be beneficial for one family, it may be considered detrimental for another, as supported by our finding that some families decline testing because they are overwhelmed. The challenge for clinicians will be in determining when the possibility of testing might best be broached with each family. While clinicians reported that some families were overwhelmed by the offer of testing, it is difficult to determine whether this was due to the offer itself or the fact that the offer had an expiry date. The suggestion raised by some clinicians of allowing parents to choose exome sequencing for hearing loss at any time (rather than as a limited time offer) may provide a solution. 

Our participants described how parents exhibited varying reactions to receiving results from exome sequencing. Most parents generally reacted positively to a genetic cause for the hearing loss, which accords with other studies that have explored health professionals’ reports of parents’ reactions to receiving results for children with other genetic conditions (Vears, 2019; Wynn, 2018). However, our participants recounted that some parents react more negatively, which may be due to the stigma associated with genetics or because a result provided unwanted confirmation of the hearing loss itself. Health professionals from other studies have suggested other reasons for negative reactions by parents, such as shock that the condition is genetic, frustration the answer did not lead to a treatment or cure, or distress that it confers a progressive course or worse prognosis than anticipated (Vears, 2019; Wynn, 2018). Similarly, mixed responses were reported by our participants in relation to negative results. While some parents expressed relief, as seen in other studies (Vears, 2019; Wynn, 2018), other parents were disappointed at not having information to make management and reproductive decisions. This disappointment has been reported by others but seemed to be more pronounced when parents had high expectations that a cause would be identified (Vears, 2019; Wynn, 2018). These findings suggest that the nature of the genetic condition being tested needs to be taken into account when preparing families to receive results.

Some of our participants raised ethical concerns relating to equity of access depending on the type of hearing loss diagnosed in the infant. While the overarching clinical study had strict eligibility criteria (permanent, bilateral congenital hearing loss of moderate, severe, or profound severity) [1], clinicians noted that testing may also be of benefit to other infants (and their families) with forms of hearing loss that did not meet these criteria (e.g., mild or unilateral). That leads us to question whether the criteria used would be ethically justifiable should sequencing be integrated into clinical care. The ethical principle of justice is relevant to answering this question. Two types of justice are particularly relevant—distributive justice (which refers to fairly allocating limited resources) and justice as equity (the notion that like cases must be treated alike) [14]. Having eligibility criteria is important from a distributive justice perspective; resources within a healthcare system are finite and challenging decisions must be made regarding how these funds are best allocated. The criteria for exome sequencing were set because those with bilateral and more severe hearing loss were predicted to have a greater chance of having a genetic cause, leading to a greater diagnostic yield. Hence, restricting the offer of exome sequencing to patients with a greater likelihood of having a cause identified may be ethically justifiable using a distributive justice lens.

However, such strict eligibility criteria may be questioned when considering the notion of justice as equity. Applying this notion of justice requires asking what the similarities and differences are between two groups, and whether these justify treating these two groups differently. It could be argued that infants with permanent, bilateral congenital hearing loss of moderate, severe, or profound severity stand to benefit more from receiving an answer for their hearing loss via exome sequencing (for example, if such infants are more likely to have syndromic forms of hearing loss and could, therefore, benefit from earlier detection and clinical management). It may also be possible that parents of these infants are more affected by their child’s hearing loss diagnosis and management and could, therefore, stand to benefit more from the opportunity to receive an answer for their child’s hearing loss. However, infants with unilateral or mild forms of hearing loss and their parents may also benefit from the opportunity to receive exome sequencing. For example, this group too may appreciate the opportunity to investigate if the cause of the hearing loss is genetic, and the reproductive planning benefits of such testing may apply equally to this group, if a genetic cause is established. On the other hand, the lower probability of finding a diagnosis also means that there is a greater chance of identifying variants of uncertain significance in these infants, which increases the workload for laboratories, the complexity of the counselling process for clinicians, and the uncertainty for families. A lower diagnostic yield also means fewer families would receive results that could be used for reproductive planning purposes. Further research assessing the diagnostic yield, as well as the clinical and personal utility of exome sequencing in infants diagnosed with unilateral and mild hearing loss, would be informative.

Another aspect relating to justice raised by some of our participants concerned the ethical justifiability of offering exome sequencing for free to some families (those who were able to sign up for the study in time) but not others (those who missed the project cut-off). This issue is, of course, inherent in all research projects as evidence for clinical benefit often needs to be generated before applications for healthcare funding can be made to include testing in routine clinical care. Nonetheless, the incorporation of exome sequencing into clinical practice will help to ensure that the offer reaches more families of infants with hearing loss who could derive benefit. In addition, health professionals may benefit from more support in helping them navigate some of these ethical challenges.

There are several factors that must be considered when drawing inferences from our results. First, although we interviewed over half of the clinicians deemed eligible to take part in this qualitative study, it is important to note that this was a self-selected sample and, therefore, there may be some views that are not represented in the data. For example, the majority of participants were recruited from one study site. Although this is partly because two-thirds of the potential participants were based at one site, it could also be due to our closer research and clinical connections with that site, which may have encouraged participation. We used a qualitative methodology, which aims to generate rich data that addresses study aims, rather than to generate data that is statistically representative of the broader population [9]. As such, our findings cannot be generalised to clinicians who offer exome sequencing for other childhood conditions. Future qualitative research conducted with clinicians who are involved with similar genetic sequencing projects in other countries (for hearing loss as well as for other paediatric conditions) would be worthwhile, given that clinicians’ views regarding exome sequencing for children will depend on various factors including the local context within which they practice.

## 5. Conclusions

Our study provides insights into clinicians’ experiences of clinical exome sequencing in infants with hearing loss. In particular, we have identified that some of the key challenges relate to when is the most appropriate time to offer testing to this cohort and who to offer it to. The findings are particularly important as much of the existing literature exploring parents’ choices and clinicians’ views regarding exome sequencing in children have been focused on older children, rather than infants. Moreover, as clinicians sought consent to exome sequencing from parents in a real-time, clinical context (rather than as a hypothetical offer), our findings have important implications for clinical practice and high relevance to health professionals working in this area. However, further research is needed to explore direct accounts of parents’ views and experiences. In particular, it will be important for future qualitative research to explore parents’ views on when such exome sequencing should be offered, as well as which children or types of hearing loss should be deemed eligible for this testing. In particular, it would be interesting to compare parents’ choices and experiences when they are allowed to choose both if and when to proceed with sequencing in this setting. It would also be worthwhile to compare and contrast the experiences of parents with and without a family history of hearing loss (including parents who also have hearing loss). Finally, more insights into how the clinicians frame the testing offer and the potential impact this has on parental decision making could be gained by observing consultations. The rich, empirical data from clinicians generated by our study will serve as a useful start point with which to compare parents’ own views and accounts of decision making regarding exome sequencing for their child’s hearing loss.

## Data Availability

Additional data are available from the corresponding author on request.

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
