# Peer review of "Clinicians’ Views and Experiences with Offering and Returning Results from Exome Sequencing to Parents of Infants with Hearing Loss"

_jcm, 2021, doi:10.3390/jcm11010035_

Round 1

Reviewer 1 Report

This was a very interesting paper on a very important topic. But there are a number of changes in the methods, results, and strengths/weaknesses section that would benefit from consideration beforehand. I made comments on the pdf which I have attached. 

Author Response

We thank the reviewer for their helpful comments, which we have addressed as follows:

Reviewer #1

Abstract:

  1. “These findings have important implications for how ES should be offered to families following the diagnosis of a child with hearing loss, particularly with regards to when testing is suggested.” Suggest “shows the complexity of”? given that the findings seem to go in contradictory directions.

We have amended this sentence to read: “These findings show the complexity involved in determining how ES should be offered to families following the diagnosis of a child with hearing loss, particularly with regards to when testing is suggested.”

Introduction:

  1. “Our study fills this gap.” Suggest “contributes to addressing”?

We have amended this sentence to read: “In order to contribute to addressing this gap, we aimed to interview health professionals who were offering exome sequencing to infants following a diagnosis of hearing loss to capture their views and experiences with the processes of offering and returning these results.”

Materials and methods:

  1. “In accordance with qualitative research methodology, participant recruitment continued until data saturation was reached (when minimal new data was generated addressing the study aims).” I'd always be cautious using this argument when there are a finite number of people in a relatively small group who can take part.

We are unsure what Reviewer 1 is concerned about here; as noted in our original manuscript, data saturation was reached despite the small number of participants.

  1. Who designed the interview schedule? Was it piloted? Any input from health professionals?

We have added a sentence to the methods section which reads: “The interview guide was developed by LN and DV and feedback was provided by the clinical study lead who is a clinical geneticist.”

  1. “The semi-structured interview guide included open-ended questions regarding the practical and ethical experiences and challenges related to the testing process as follows: a) clinicians’ views on whether and when exome sequencing should be offered to parents of children with hearing loss; b) clinicians’ accounts of parents’ reasons for and against exome sequencing for their child’s hearing loss; and c) clinicians’ experiences returning hearing-loss-related exome sequencing results to parents.” Were experiences of offering the sequencing gathered? From the text so far i thought that was part of the focus, but it's not clear from these areas that it was explored. Perhaps clarify?

We have amended this section, which now reads “a) clinicians’ views on whether and when exome sequencing should be offered to parents of children with hearing loss; b) clinicians’ accounts of parents’ reasons for and against exome sequencing for their child’s hearing loss; and c) clinicians’ experiences offering hearing-loss-related exome sequencing and returning results to parents.”

  1. in person? on phone? In work setting? I've seen this information is in later section. I would put it in methods as it has important methodological impact.

We have moved the information from the results to the methods, which now reads “Interviews were conducted by LN either in person or by telephone, depending on the location and preference of the participant. In person interviews were conducted at the participant’s workplace, either in a meeting room or in the participant’s office”

  1. How did you know when coding had finished?

We have added a sentence to the methods: “Coding continued iteratively until all data relevant to the research question had been coded into categories and subcategories.”

Results:

  1. I realise response rates are not relevant here, but given the slight issue with the saturation argument for the sample, i was interested in how these participant numbers fitted with the overall possible participants for this study.

We have added to the results section, which now reads: “A total of 12 clinicians participated, including seven (58%) genetic counsellors, four (33%) clinical geneticists, and one (8%) paediatrician; this was more than half of the 21 clinicians eligible to participate in the study.”

  1. Was there a difference in role as to whether they were in person or on telephone?

There was no difference in role as to where the interviews were conducted. It was based more on convenience of which hospital site the participant was based and their availability. We have added this into the methods as follows: “Interviews were conducted by LN either in person or by telephone, depending on the location and preference of the participant.”

  1. “While clinicians were recruited from both study sites, the majority (10/12, 83%) worked at one site.” Why was this?

There are a couple of possible reasons for this. The main reason is that 2/3 of the eligible participants were based at one site so we had a larger pool to sample from. Our team also has stronger connections with that site. We have added these to the limitations as follows: “For example, the majority of participants were recruited from one study site. Although this is partly because two-thirds of the potential participants were based at one site, it could also be due to our closer research and clinical connections with that site, which may have encouraged participation.”  

  1. “Three main content categories were generated from the interview data: 1) clinicians’ views on offering exome sequencing to parents of children with hearing loss; 2) clinicians’ accounts of parents’ decision making regarding exome sequencing for their child’s hearing loss; and 3) clinicians’ experiences returning diagnostic exome sequencing results to parents.” This is slightly problematic as these map directly onto what was stated as being the areas of your interview schedule. When this is the case it is always hard to answer convincingly that the categories are data generated rather than being chosen a priori. There is no problem in the categories being decided a priori and at times this is very useful in health services research, but if this was the case, make that clear in your analysis section. At the moment this doesn't seem convincing (though i'm sure the work is fine).

We agree that this concept of results categories mapping closely with the interview guide is problematic when using thematic analysis, when the goal is to develop themes that should resemble overarching concepts. However, as we state in the methods, we used inductive content analysis, in which the data are coded much more closely to the text (rather than abstracted) and therefore the content categories often do resemble the questions asked.

  1. “The diagnosis of certain genetic syndromes will lead you down a different treatment pathway and allow you to avoid other investigations.” iers something about their role OR within the analysis or narrative sections letting us know whether there were differences in roles. I say this as in the abstract it was made clear that views were fairly polarised. As such it's important to understand why as otherwise it's hard to make suggestions for practice

If we have understood Reviewer 1 correctly, we think they are asking whether differing opinions of the participants could be due to their profession/role. Interestingly there was no clear pattern with clinician type, study site or years of experience so it seems this is due to individual differences in clinician opinion which may be based on their experiences with different families. We have added a line into the discussion to reflect this as follows: “Interestingly there was no clear pattern with clinician type, study site or years of experience as to whether participants were more or less in favour of offering exome sequencing for hearing loss.”

  1. “Some clinicians also expressed concerns about offering exome sequencing for paedi-atric hearing loss.” Ok can you tell us anything about these participants as to why they have different views from those above?

As per the previous response, there was no clear pattern with clinician type, study site or years of experience so it seems this is due to individual differences in clinician opinion which may be based on their experiences with different families. We have added a line into the discussion to reflect this as follows: “Interestingly there was no clear pattern with clinician type, study site or years of experience as to whether participants were more or less in favour of offering exome sequencing for hearing loss.”

  1. “However, most clinicians felt that exome sequencing should be offered to parents at a later date, often because the children were very young infants and parents were still coming to terms with their child’s hearing loss diagnosis.” it sounds like the difference in these views then are when they are thinking about different families rather than different clinicians having different views? As this has implications on how this would be offered, it would be great to get that insight into this from your study.

Yes, this appears to be the case although it was not necessarily said this explicitly by the participants (hence why we have not included it in the results section). We do address this point in the discussion however, as follows: “The appropriate timing to offer exome sequencing for hearing loss will likely depend on the particular family and their circumstances. While offering early exome sequencing may be beneficial for one family, it may be detrimental for another, as supported by our finding that some families decline testing because they are overwhelmed.”

  1. “So I think people [study clinicians] felt a bit uncomfortable about that, that it was all happening at the wrong time...not the genetic testing for the cause for the deafness, but the offer, and the idea that they were being invited for two or three projects...simultaneously at a point when they’re still coming to grips with the diagnosis in the child.” Are they referring to themselves/others in peer group? or guessing? It would be great to have something in the narrative to make this clear. Always an issue when someone is reporting how someone else feels, so it's good to have some insight into their ability to make that assumption

This clinician supervises other clinicians who were also offering testing to parents in the clinical study, so this clinician is referring to their colleagues in this quote. We have made this clearer as follows: “Families are also being invited to participate in other studies relating to hearing loss at this time, and some clinicians discussed the discomfort they and their colleagues experienced offering the test.”

  1. “Some clinicians felt that the decision regarding exome sequencing has important im- plications, and therefore parents should be given more time to decide. Some clinicians felt that there should be no time limit on the offer, and parents should be able to opt in to receive exome sequencing at any time:” sorry in which case fine, but very helpful for the reader to know. Otherwise we have a range of differing views and don't know what to do with them.

If we have understood correctly, Reviewer 1 is again questioning why participants had varying views about whether and how exome sequencing should be offered in that context. Again, no clear pattern with clinician type, study site or years of experience so it seems this is due to individual differences in clinician opinion based on their experiences with different families. We should also note that these two views are not mutually exclusive; parents can be given time to decide AND there could be no time limit on the offer. As for not knowing what to do with the differing views, in our view they help to illustrate the nature and complexity of the issue.

  1. “Most clinicians reported that some parents also accepted exome sequencing for their child for reasons relating to reproductive planning. One clinician noted that some parents wanted to be better prepared if they had another child with hearing loss:” Can you just make this a little bit clearer as we have a "most clinicians" followed by a "one clinician" for what look like they could be similar ideas (i think i know why they are different, but make this explicit why you are highlighting this one clinician).

We have made this clearer as follows: “For example, some parents wanted to be better prepared if they had another child with hearing loss as noted by this clinician.”

  1. “Conversely, several clinicians stated that other parents wanted to avoid having another child with hearing loss: “ don't really need this data - it's not adding anything

We actually think including this as another reproductive implication of exome sequencing is important. As such we have elected to keep it but have tried to make the relevance clearer as follows: “In contrast, other parents wanted to use the results in order to avoid having another child with hearing loss:

  1. “I just remember them saying, ‘I don’t mind whether my baby’s hearing loss is because of a genetic change or not, it’s not going to change anything for us or our family.’”just make clearer whether this is an example of a range of responses parents gave OR really a one off.

To show that this was not just a ‘one-off’ finding, we have changed the proceeding sentence to read: “First, clinicians reported that some parents declined participation as they felt that the exome sequencing results would not change anything:”

  1. “There’s so much support, programs, intervention related to hearing loss that the families just felt run off their feet and overwhelmed by all the services...they were like, ‘I can’t introduce another health service into the picture.’” This seems to be linked to your timing idea above... i would make that link explicit

We have added a sentence in the discussion, which now reads: “This finding was supported by participants’ reports that some parents declined the offer because of how overwhelmed they were at the time.”

  1. “Some parents felt not receiving a diagnosis for their child’s hearing loss would be too disappointing.” Can you explain this for readers? It's the first time we've come across the idea that families could go through this and not get an answer.

We state in the introduction that “testing identified a genetic cause of hearing loss in 56% (59/106) of the infants who participated in the study”, which suggests that a proportion of participants do not receive a causative finding. We have added to this sentence to make that clearer, which now reads “testing identified a genetic cause of hearing loss in 56% (59/106) of the infants who participated in the study, leaving 44% for whom a causative variant was not identified.” We have also amended the sentence in the results to read: “Some parents felt that, given the possibility a diagnosis for their child’s hearing loss would not be identified, accepting testing might lead to disappointment.”

  1. “Others either did not want to be involved in research or had concerns about the potential implications of the findings, such as the impact of exome sequencing results on their child’s future ability to obtain insurance or concern it would reveal the child’s father was not their genetic father.” This reads like a lot of really important reasons which it would be crucially important to understand if we're going to a. offer a fair/accessible service b. offer a service with no harm. Personally, i would really advise making this section longer and adding in data. It seems a bit unbalanced with the length of some sections above.

We have added another quote to this section to help illustrate this point and expand the section: “I think at the moment I’d just say it’s a mixture of different factors, which range from they may already have a genetic diagnosis, they may have decided the condition’s not genetic. They may just not want to be involved in research, or be concerned about insurance or other implications. So we’ve got a list, but there’s no sort of single factor that’s repeating, it’s basically a whole lot of different things.” Clinician 1

  1. “One clinician highlighted an experience where the parents had been warned of a likely syndromic diagnosis and had time to come to terms with this possibility: “I had a particular family where the geneticist involved had sort of guessed a syndromic diagnosis before we got it, so they were already in a frame of mind that that’s what their baby had, and so we came back to them and said ‘yep, that doctor was right...we’ve confirmed now that this is the diagnosis for your baby’. So even though that was probably a pretty devastating kind of diagnosis for the family, they had some time to adjust to that before we actually came back with the result, which was interesting.”” i.e. something has caused you to group this data together

We have decided that this point did not really fit well with this section and have elected to remove it.

  1. For parents who did not receive an explanation for their child’s hearing loss, clinicians reported mixed reactions. Many parents who did not receive an explanation reacted quite well and were particularly relieved that no syndromic diagnosis was found: “You have families where you think that they’re going to be devastated that you didn’t find something because they really, really, really wanted the answer, but then actually they managed it quite well and they’re very comfortable with where things are at.” (Clinician 5) “As much as we didn’t find a diagnosis at all, a lot of families were relieved that we didn’t find a syndromic diagnosis or something like that.” I don't think you need both bits of data

We have removed one of the quotes as suggested.

Discussion:

  1. “This is the first published study exploring clinicians’ views and accounts of parental decision making regarding exome sequencing for hearing loss in infancy.” This sounds different from what the original aim was phrased as.

We have amended this to read: “This is the first published study exploring clinicians’ views and experiences with offering and returning results from exome sequencing for hearing loss in infancy”.

  1. “Parents may already be overwhelmed by either the fact they have a newborn, particularly in the case of first-time parents, or the recent hearing loss diagnosis.” Did you data give us any insight into whether this was affecting first time parents more?

This is an interesting point, but it was not raised by any of our participants so it was merely speculation. As such, we have adjusted the sentence, which now reads “Parents may already be overwhelmed by either the fact they have a newborn or the recent hearing loss diagnosis.”

  1. “However, concerns about offering such testing were also raised by some clinicians, particularly in the context of isolated (non-syndromic) hearing loss, for which the clinical benefits of offering such testing may be less clear.” When you present two sides of the argument like this, it would be really helpful to say WHY - so some of this seemed to be around costs/expectations raised by study and availability outside of that/morality of testing for deafness.... try and tease it out so we can see a way forward.
  2. “While offering early exome sequencing may be beneficial for one family, it may be detrimental for another.” As shown in the data where families turned down due to overwhelm.

We have added to the existing sentence, which now reads “While offering early exome sequencing may be beneficial for one family, it may be considered detrimental for another, as supported by our finding that some families decline testing because they are overwhelmed.”

  1. “Future research could compare parents’ choices and experiences when they are allowed to choose both if and when to proceed with sequencing in this setting.” I liked this, but it read like well developed narrative for the results section. In a discussion section i'd want to see more of lining it up with existing literature or showing how there are novel insights.

The statement Reviewer 1 refers to is not a research finding (and therefore would not belong in the results). It is a comment about an important avenue that future research could explore. To reduce confusion, we have moved this to the conclusion where we discuss other future research avenues.

We have also added another paragraph which compares our findings to those of others regarding experiences returning results and the bibliography has been updated accordingly: “Our participants described how parents exhibited varying reactions to receiving results from exome sequencing.  Most parents generally reacted positively to a genetic cause for the hearing loss, which accords with other studies that have explored health professionals’ reports of parents’ reactions to receiving results for children with other genetic conditions (Vears 2019; Wynn 2018). However, our participants recounted that some parents react more negatively, which may be due to the stigma associated with genetics or because a result provided unwanted confirmation of the hearing loss itself. Health professionals from other studies have suggested other reasons for negative reactions by parents, such as shock that the condition is genetic, frustration the answer did not lead to a treatment or cure, or distress that it confers a progressive course or worse prognosis than anticipated (Vears 2019; Wynn 2018). Similarly, mixed responses were reported by our participants in relation to negative results. While some parents express relief, as seen in other studies (Vears 2019; Wynn 2018), other parents were disappointed at not having information to make management and reproductive decisions. This disappointment has been reported by others but seemed to be more pronounced when parents had high expectations that a cause would be identified (Vears 2019; Wynn 2018). These findings suggest that the nature of the genetic condition being tested needs to be taken into account when preparing families to receive results.”

  1. “Hence, restricting the offer of exome sequencing to patients with a greater likelihood of having a cause identified may be ethically justifiable using a distributive justice lens.” This sounds like a good argument. It made me wonder whether the health professionals in the sample knew this? Was it a case of lack of knowing this OR was it a case of they knew this and yet in practice this still felt hard? I think this has important implications for the paper in terms of - do we need to make these things clearer when putting clinicians in this setting OR is it something people are going to struggle with anyway and perhaps it's more about recognising that and support?

Our hypothesis is that only those involved in the study design may have aware of this argument, not the rest of our participants. However, we do not have any data to support this so feel it would be inappropriate to speculate on this in the manuscript. However, we have added a sentence in the discussion at the end of the section on distributive justice as follows: “In addition, health professionals may benefit from more support in helping them navigate some of these ethical challenges.”

  1. Nonetheless, incorporation of exome sequencing into clinical practice will help to ensure that the offer reaches more families of infants with hearing loss who could derive benefit.” I accept all of this, but what is the learning we can take from this in terms of clinicians offering this testing? It would be great to bring this back here.

We have added a sentence to the conclusion to highlight the key learnings: “In particular, we have identified that some of the key challenges relate to when is the most appropriate time to offer testing to this cohort and who to offer it to.”

  1. “First, our total sample size was relatively small, although we interviewed over half of the clinicians deemed eligible to take part in this qualitative study.” On what basis? Reference this to back up that it is seen as small.... but also be careful with this, it's going down a positivistic, quantitative route - be clear what the issue is for the qual sample being this size. This would be useful information earlier, but again be careful how you use it. It's context, but it's not justification for robustness of sample.
  2. “While our sample size may be considered small, we achieved data saturation and heterogeneity through recruiting at least one clinician from each of the clinical specialties involved in the overarching clinical study (genetic counselling, clinical genetics, and paediatrics) and each of the study sites.” Judged by who? On what grounds? Back this up - how do you know you achieved this? Rather than focusing on sample size, i would focus on this point as this is how you set out your sample in your design. However, saturation is a contested concept. I would weave that argument in here rather than focus on small sample size which sounds odd in this setting.

To address points 31 and 32, we have removed the comment about the sample size as Reviewer 1 correctly notes that this is not necessary based on our methodological approach. We have also removed the statement about data saturation from the limitations and added a little more detail to the methods section, which now reads: “In accordance with qualitative research methodology, participant recruitment continued until data saturation was reached (when minimal new data was generated addressing the study aims) and we had achieved sufficient sample heterogeneity by recruiting at least one clinician from each of the clinical specialties involved in the overarching clinical study (genetic counselling, clinical genetics, and paediatrics) and each of the study sites.”

  1. “However, our findings cannot be generalised to clinicians who offer exome sequencing for other childhood conditions.” No, but qual doesn't seek to generalise. Rather we provide context to enable transferability. That's why i was urging for more context above.

Reviewer 1 makes a good point. At the same time, we thought the findings not being generalizable is important to include as many readers will not be familiar with qualitative research. As such we have reframed this paragraph, which now reads: “There are several factors which must be considered when drawing inferences from our results. First, although we interviewed over half of the clinicians deemed eligible to take part in this qualitative study it is important to note that this was a self-selected sample and therefore there may be some views that are not represented in the data. We used a qualitative methodology, which aims to generate rich data that addresses study aims, rather than to generate data that is statistically representative of the broader population [9]. As such, our findings cannot be generalised to clinicians who offer exome sequencing for other childhood conditions. Future qualitative research conducted with clinicians who are involved with similar genetic sequencing projects in other countries (for hearing loss as well as for other paediatric conditions) would be worthwhile, given that clinicians’ views regarding exome sequencing for children will depend on various factors including the local context within which they practice.”

  1. “Critically, our study did not directly capture parents’ views and experiences of being asked to decide about exome sequencing for their child’s hearing loss.”Agree, but it didn't seek to. I would like to see more here about the strengths and weaknesses of what happened in your study.

This is a good point. We have moved the points about a need for further research into parents’ views into the conclusion accordingly. We have also restructured and added to the limitations section as described previously. The strengths are captured in the conclusion: “The findings are particularly important as much of the existing literature exploring parents’ choices and clinicians’ views regarding exome sequencing in children has been focused on older children, rather than infants. Moreover, as clinicians sought consent to exome sequencing from parents in a real-time, clinical context (rather than as a hypothetical offer), our findings have important implications for clinical practice and high relevance to health professionals working in this area.”

  1. “Moreover, as clinicians sought consent to exome sequencing from parents in a real-time, clinical context (rather than as a hypothetical offer), our findings have important implications for clinical practice and high relevance to health professionals working in this area.” Also clinician's views likely to influence parents.

This is another interesting point, although obviously we do not have the data to comment on this. We have added it as something for further research: “Finally, more insights into how the clinicians frame the testing offer and the potential impact this has on parental decision-making could be gained by observing consultations.”

Reviewer 2 Report

This article deals with a very important topic, as exome sequencing is increasingly used in clinical practice. Even though it is only a qualitative approach, the text gives a good overview of the pros and cons of this topic for children with congenital hearing disorders.

Introduction:

Line 63: it would be interesting to know, why exome sequencing has the potential to be integrated into routine clinical care

Line 69: The aims of the study should be presented in more detail

Results:

Table 1 gives no further information. The years of working in current profession can be described in one sentence and should be discussed if they influence the opinions of the interview partners. The results are interesting to read with the illustrative quotes and covers all the important points in the three categories.

Discussion:

In my opinion, the results found should be compared with the literature in a discussion. Even if this is the first study on exome sequencing in congenital hearing disorders, there should be experiences with exome sequencing in other diseases to discuss. In its current form, the discussion mainly repeats the results already described and therefore definitely needs to be revised. Accordingly, the bibliography should also be adapted.

Additionally, I think that lines 376-413 do not necessarily need to be discussed because they concern the study design and only address one detail of the interviews.

Author Response

We thank the reviewer for their helpful comments, which we have addressed as follows:

Reviewer #2

Introduction:

  1. Line 63: it would be interesting to know, why exome sequencing has the potential to be integrated into routine clinical care.

We have added to this sentence which now reads: “The study by Downie et al. showed that, based on the high diagnostic yield and clinical implications, exome sequencing following the identification of infants with hearing loss has the potential to be integrated into routine clinical care.”

  1. Line 69: The aims of the study should be presented in more detail.

We have added to the end of the introduction which now reads: “In order to contribute to addressing this gap, we aimed to interview health professionals who were offering exome sequencing to infants following a diagnosis of hearing loss to capture their views and experiences with the processes of offering and returning these results.”

Results:

  1. Table 1 gives no further information. The years of working in current profession can be described in one sentence and should be discussed if they influence the opinions of the interview partners. The results are interesting to read with the illustrative quotes and covers all the important points in the three categories.

We have removed Table 1 from the text and instead add the following to the text: “Clinicians varied in terms of how many years they had worked in their current profession; experience ranged between trainee level (n=1) and >20 years (n=1) with the majority of participants having from 1-4 years (n=4) or 5-9 years (n=3) experience.” Years’ experience did not appear to influence views of participants, which has been added to the discussion: “Interestingly there was no clear pattern with clinician type, study site or years of experience as to whether participants were more or less in favour of offering exome sequencing for hearing loss.”

Discussion:

  1. In my opinion, the results found should be compared with the literature in a discussion. Even if this is the first study on exome sequencing in congenital hearing disorders, there should be experiences with exome sequencing in other diseases to discuss. In its current form, the discussion mainly repeats the results already described and therefore definitely needs to be revised. Accordingly, the bibliography should also be adapted.

We thank Reviewer 2 for alerting us to this oversight. We have added another paragraph which compares our findings to those of others regarding experiences returning results and the bibliography has been updated accordingly: “Our participants described how parents exhibited varying reactions to receiving results from exome sequencing.  Most parents generally reacted positively to a genetic cause for the hearing loss, which accords with other studies that have explored health professionals’ reports of parents’ reactions to receiving results for children with other genetic conditions (Vears 2019; Wynn 2018). However, our participants recounted that some parents react more negatively, which may be due to the stigma associated with genetics or because a result provided unwanted confirmation of the hearing loss itself. Health professionals from other studies have suggested other reasons for negative reactions by parents, such as shock that the condition is genetic, frustration the answer did not lead to a treatment or cure, or distress that it confers a progressive course or worse prognosis than anticipated (Vears 2019; Wynn 2018). Similarly, mixed responses were reported by our participants in relation to negative results. While some parents express relief, as seen in other studies (Vears 2019; Wynn 2018), other parents were disappointed at not having information to make management and reproductive decisions. This disappointment has been reported by others but seemed to be more pronounced when parents had high expectations that a cause would be identified (Vears 2019; Wynn 2018). These findings suggest that the nature of the genetic condition being tested needs to be taken into account when preparing families to receive results.”

  1. Additionally, I think that lines 376-413 do not necessarily need to be discussed because they concern the study design and only address one detail of the interviews.

We thank Reviewer 2 for providing their perspective. However, as we state, we are not just referring to the issue of equity in relation to inclusion in the study, we are also referring to equity in access should genomic sequencing become a routine part of clinical care. As such, we have elected to retain this section because we believe it is highly relevant to the discussion.

Round 2

Reviewer 2 Report

The article is better and two papers are included in the discussion. In my opinion it can be published now